# A Symmetric Matrix-Aided MIMO to Improve Reliability for Maritime Visible Light Communications

**You Zhang** [1,†] **, Yan Feng** [1,*,†] **, Lin Zhang** [1,2,†] **and Zhiqiang Wu** [1,3,†]

1 School of Information Science and Technology, Tibet University, Lhasa 850000, China
2 School of Electronics and Information Technology, Sun Yat-sen University, Guangzhou 510006, China
3 Department of Electrical Engineering, Wright State University, Dayton, OH 45435, USA
* Correspondence: fy4528@163.com
† These authors contributed equally to this work.

**Abstract:** Turbulence in natural environments affects the reliability of communication. In this paper, we propose a symmetric matrix-assisted multiple-input multiple-output (MIMO) maritime visible light communication (VLC) system to confront turbulence and improve reliability. In our design, by exploiting the repeatability of elements in the symmetric matrix, the matrix can expand the Euclidean distance between signals and effectively restrains the interference between signals, thus improving the reliability of maritime VLC systems. In addition, we derive the theoretical symbol error rate (SER) of the proposed scheme. Then, simulation results are provided to validate the theoretical SER. In addition, the SER performances of the proposed system are compared with the three benchmark schemes of repetitive coding (RC), spatial modulation (SM), and spatial multiplexing (SMP), and the transmission performances in VLC systems with different link distances, water qualities, and wind speeds are also investigated. The proposed symmetric matrix-assisted MIMO maritime VLC system can combat interferences effectively and enhance reliability performance.

**Keywords:** multiple-input multiple-output (MIMO); visible light communication (VLC); symmetric matrix; reliability

## 1. Introduction

In the past few years, with the widespread application of optical communication, there has been a great deal of interest in visible light communication (VLC). Compared with other wireless communication technologies, visible light communication (VLC) is characterized by rich spectrum resources, less electromagnetic interference, and high security [1]. In recent years, along with the increasing explorations in oceans, VLC systems have been applied to provide information transmission services for maritime communication [2–5].

However, for maritime channels, the conditions are complex due to the dramatic variations of the underwater transmission medium, thereby leading to reliability performance degradation. In order to improve the reliability of underwater VLC (UVLC) systems, researchers have proposed to apply the multiple-input multiple-output (MIMO) technology to exploit the spatial diversity to lower the bit error rate [6–12]. With the aid of the MIMO scheme, maritime VLC systems can combat the scattering and turbulence in the channels. As mentioned in [13], turbulence can be divided into weak turbulence and moderate to strong turbulence, and the log-normal model is used to describe weak turbulence, while the gamma-gamma model is used to describe moderate to strong turbulence.

Although the MIMO-VLC systems can exploit the spatial diversity gain to improve performance, inter-channel interferences still exist due to the imperfect MIMO transmissions. Moreover, the synchronization circuit is complex to be implemented. To address these issues, Refs. [14,15] proposed the spatial multiplexing (SMP) scheme and spatial modulation (SM) scheme to enhance the transmission performance.

Different from the existing works, in this paper, we propose a symmetric matrix-assisted MIMO transmission scheme. By exploiting the repeatability in both time and space domains of the elements in the symmetric matrix, the transmission reliability can be improved thanks to the enlarged minimum Euclidean distance. In this design, after the user information is converted from serial to parallel (S/P), the generated bits stream is modulated into the pulse amplitude modulation (PAM) symbol and then transformed by the symmetric matrix module. Thereafter, the double singular value decomposition (SVD) method is used to suppress the interferences between different users. Subsequently, we derive the theoretical symbol error rate (SER) expression of the system proposed in this paper. Then, the simulation results are carried out to verify the accuracy of the derivations. In addition, we compare the performance between the proposed scheme and the benchmark schemes including repetitive coding (RC), spatial modulation (SM), and spatial multiplexing (SMP).

Briefly, the major contributions include:

1. We propose to use a symmetric matrix-assisted MIMO transmission to improve the reliability of maritime VLC systems. By exploiting the repeatability of the elements in the symmetric matrix, the minimum Euclidean distance can be enlarged.
2. We use the double SVD to suppress the interferences between the users. Then we apply the scheme in maritime VLC systems.
3. We derive the theoretical SER expression for the proposed scheme. Then simulation results are provided to verify the effectiveness of the scheme proposed in this paper.

The remainder of this paper will proceed with a description of the details of the maritime MIMO-VLC system characteristics in Section 2. A discussion of the symmetric matrix-aided MIMO-VLC system is given in Section 3. Then, Section 4 presents the simulation results and Section 5 concludes the findings.

## 2. Maritime MIMO-VLC Channel Models

In this section, we will describe three channel models. Referring to [16,17], we assume that the deviation caused by the small displacement of the receiver or transmitter or the aperture reception is negligible.

For the atmospheric and underwater channels, denote $h_p$ and $h_t$ as the path loss coefficient and channel fading coefficient, respectively. Then the channel coefficient $h$ can be expressed as [18]:

$$h = h_p \cdot h_t. \tag{1}$$

For the underwater channel, we use the Monte Carlo (MC) method to approximate $h_p$ [19]. For the atmospheric channel, the path loss coefficient $h_p$ depends on the geometric loss led by the receiver aperture area, transmitter divergence angle and the atmospheric turbulence caused by the absorption and scattering [20]. The $h_p$ can be received as follows:

$$h_p = \frac{D}{\pi(\phi L_{au}/2)^2} \exp(-\beta_v L_{au}), \tag{2}$$

where $D$ is the area of the receiver's aperture, $L_{au}$ means the propagation distance, $\phi$ represents the divergence angle, and $\beta_v$ is on behalf of the atmospheric extinction coefficient.

The channel response $h_t$ is related to turbulence. The double gamma model is used to describe the moderate to strong turbulence, while the turbulence intensity is weak, the log-normal model is applied [13,20]. The log-irradiance variance $\sigma_I^2$ is used to reshape the turbulence intensity, which can be [20,21]:

$$\sigma_I^2 = 1.23 C_n^2 k^{7/6} L_{au}^{11/6}, \tag{3}$$

where $C_n^2$ is the turbulence strength parameter, whose value ranges from $10^{-17}$ to $10^{-12}$. $k = 2\pi/\lambda$ is the phase constant.

To describe the statistical characteristics of the weak turbulence, with reference to [22], we will use the log-normal distribution, which is the most widely used model of irradiance probability density function(PDF). Namely, the PDF of $h_{t,w}$ can be expressed as [22]:

$$f(h_{t,w}) = \frac{1}{\sqrt{2\pi}\sigma_I h_{t,w}} \exp\left(-\frac{(\ln(h_{t,w}) + \sigma_I^2/2)^2}{2\sigma_I^2}\right). \tag{4}$$

On the other hand, in the case of moderate to strong turbulence, the log-normal distribution data show a significant deviation from the measured data, because the log-normal distribution underestimates the role of the tail. In the recent research work, a gamma-gamma probability density function model is proposed in [23], which is a two-parameter distribution derived based on the theory of double random scintillation. Let $h_{t,s}$ denote the channel condition corresponding to the moderate to strong turbulence, which can be modeled as a stationary stochastic process characterized by a gamma-gamma distribution [23]:

$$f(h_{t,s}) = \frac{2(\xi\gamma)^{(\xi+\gamma)/2}}{\Gamma(\xi)\Gamma(\gamma)} h_{t,s}^{(\xi+\gamma)/2-1} B_{\xi-\gamma}(2\sqrt{\xi\gamma h_{t,s}}), \tag{5}$$

where $\Gamma(\cdot)$ represents the gamma function, $B_{\xi-\gamma}(\cdot)$ represents the modified Bessel function, and the order is $\xi - \gamma$ [18]. Here, $\xi$ and $\gamma$ are the effective number of small-scale and large-scale eddies of the turbulent environment, where $\xi = \{\exp[\frac{0.49\sigma_R^2}{(1+1.11\sigma_R^{12/5})^{7/6}}] - 1\}^{-1}$, $\gamma = \{\exp[\frac{0.51\sigma_R^2}{(1+1.69\sigma_R^{12/5})^{5/6}}] - 1\}^{-1}$ [18].

The air–water interface channel is more sophisticated because the medium on both sides is different and the reflection or refraction and other phenomena in the propagation process will happen, resulting in distortion [24]. Here we use the Monte Carlo method to describe the air–water interface channel regime [25]. When the light propagates the interface, the direction of light is determined by the azimuth angle $\theta_a$ and the polar angle $\theta_p$, if wind is present. The $\theta_a$ changes in $[0, 2\pi]$, while $\theta_p$ is represented by [24]:

$$f(\theta_p) = \frac{2}{\sigma^2} \exp\left(\frac{-\tan^2\theta_p}{\sigma^2}\right) \tan\theta_p \sec^2\theta_p, \tag{6}$$

where $\sigma^2$ is determined by the wind speed $V$ showed by $\sigma^2 = 0.003 + 0.00512V$ [24]. With the expression (6), the direction of the $\overrightarrow{\mathbf{n}}$ might change with the wind speed.

## 3. Symmetric Matrix-Aided MIMO-VLC System

In this section, we will introduce the MIMO-VLC system and propose the symmetric matrix design. Then theoretical performance analysis is also provided.

### 3.1. MIMO Multiuser Transmitter

In the maritime MIMO-VLC system, from Figure 1, let $N_t$ denote the number of transmitting apertures in the transmitter, these transmit apertures will transmit signals to $K$ users. In this paper, let the user $j$ as an example, the user $j$ has $N_{r,j}$ receiving apertures, the number of receiving apertures is $N_r = \sum_{j=1}^{j=K} N_{r,j}$.

Let $\mathbf{L}_j$ represent the precoding matrix from the double singular value decomposition (SVD) operations. The objective of the double SVD is to generate the precoding matrix to help eliminate the multi-user interferences (MUIs) using the subchannels decomposed by SVD orthogonal to each other. Then we use $\mathbf{x}_j$ to denote the signal of the user $j$ encoded by a symmetric matrix. After the signal $\mathbf{x}_j$ has been precoded, the signal $\widetilde{\mathbf{x}}_j$ is obtained by $\widetilde{\mathbf{x}}_j = \mathbf{L}_j \mathbf{x}_j$.

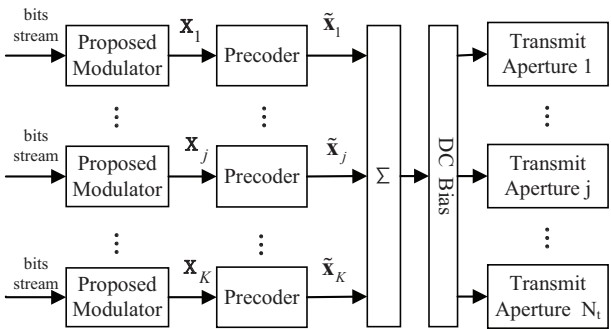

**Figure 1.** Transmitter structure for system.

To derive the precoding matrix $\mathbf{L}_j$, we first apply SVD to the $((N_r - N_{r,j}) \times N_t)$ complementary channel matrix $\widetilde{\mathbf{H}}_j$ [15]:

$$\widetilde{\mathbf{H}}_j = \widetilde{\mathbf{U}}_j \widetilde{\mathbf{\Lambda}}_j [\widetilde{\mathbf{V}}_j^{(1)} | \widetilde{\mathbf{V}}_j^{(0)}]^T, \tag{7}$$

where the $((N_r - N_{r,j}) \times (N_r - N_{r,j}))$ matrix $\widetilde{\mathbf{U}}_j$ has the left singular vectors, and the $((N_r - N_{r,j}) \times N_t)$ matrix $\widetilde{\mathbf{\Lambda}}_j$ is the diagonal matrix which is formed by ordered singular values. Let $\widetilde{P}_j$ denote the rank of the $\widetilde{\mathbf{H}}_j$, then the $(N_t \times \widetilde{P}_j)$ matrix $\widetilde{\mathbf{V}}_j^{(1)}$ holds the first $\widetilde{P}_j$ right singular vectors. In addition, the $(N_t \times (N_t - \widetilde{P}_j))$ matrix $\widetilde{\mathbf{V}}_j^{(0)}$ forms the orthogonal basis for the null space of the $\widetilde{\mathbf{H}}_j$. When the channel matrix has row full rank and the $N_t = N_r$, $N_t - \widetilde{P}_j$ is equal to the $N_{r,j}$ [15]. Then we use the $\widetilde{\mathbf{V}}_j^{(0)}$ obtained from Equation (7) to multiply the channel matrix $\mathbf{H}_j$ to generate the equivalent channel matrix $\overline{\mathbf{H}}_j$:

$$\overline{\mathbf{H}}_j = \mathbf{H}_j \widetilde{\mathbf{V}}_j^{(0)}. \tag{8}$$

Then we conduct the second SVD operation on Equation (8) [15]:

$$\overline{\mathbf{H}}_j = \mathbf{H}_j \widetilde{\mathbf{V}}_j^{(0)} = \mathbf{U}_j \mathbf{\Lambda}_j \mathbf{V}_j^T, \tag{9}$$

where the $(N_{r,j} \times N_{r,j})$ matrix $\mathbf{\Lambda}$ is a diagonal matrix that contains the singular values of the equivalent channel matrix $\overline{\mathbf{H}}_j$ and the $(N_{r,j} \times N_{r,j})$ matrix $\mathbf{U}_j$ is used in the receiver to decode. The $(N_{r,j} \times N_{r,j})$ matrix $\mathbf{V}_j$ is the right singular vector when the equivalent channel matrix $\overline{\mathbf{H}}_j$ is of the full rank. Moreover, we can notice that the precoding matrix $\mathbf{L}_j = \widetilde{\mathbf{V}}_j^{(0)} \mathbf{V}_j$, which falls into the null space of the $\widetilde{\mathbf{H}}_j$ to eliminate the multi-user interferences (MUIs) [15].

Then the transmitted vector $\mathbf{s}$ can obtained as [26]

$$\mathbf{s} = \sum_{j=1}^{K} (\widetilde{\mathbf{x}}_j) + \mathbf{d} = \sum_{j=1}^{K} (\mathbf{L}_j \mathbf{x}_j) + \mathbf{d}, \tag{10}$$

where $\mathbf{d}$ is the direct current (DC) bias which is added to make sure the transmitted signal is non-negative [18]. Then the signals $\mathbf{s}$ are fed into the transmit apertures.

*3.2. Symmetric Matrix-Aided Modulator*

According to the pulse amplitude modulation (PAM) principle [18], we embed the symmetric matrix after the PAM modulation. As shown in Figure 2, there are three modules: S/P, the PAM mapping, and the symmetric matrix. Here, S/P represents the serial to parallel conversion, and the PAM mapping maps each data stream to a PAM symbol, $u_1, u_2, \ldots, u_p \in \mathbb{U}$, where $\mathbb{U}$ represents the M-PAM constellation. We use the symmetric matrix $\mathbf{S}$ to encode the vector $\mathbf{u}$, $\mathbf{S}$ is as follows:

$$\mathbf{S}_{2\times 2}(a,b) = \begin{bmatrix} b & a \\ a & -b \end{bmatrix},$$  (11)

where the dimension of **S** is $2 \times 2$. The elements $a$ and $b$ must be real numbers in the symmetric matrix. In the considered MIMO-VLC system, we assume that the transmit power of the transmitter keeps constant. Thus $a$ and $b$ should satisfy $\sqrt{a^2 + b^2} = 1$.

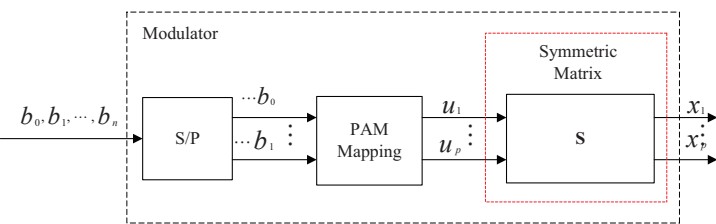

**Figure 2.** The symmetric matrix-aided modulator.

As shown in Figure 2, the PAM vector $\mathbf{u} = [u_1, u_2, \ldots, u_p]^T$ will be modulated by the symmetric matrix **S**, then the modulated signal **x** will be attained as:

$$\mathbf{x} = \mathbf{S} \cdot \mathbf{u},$$  (12)

where the **x** is substituted for the $\mathbf{x}_j$ of the $j$th user. After that process, the attained vector will be sent to the precoder.

Next, we present the symmetric matrix **S** expression. Define a error vector $\check{\mathbf{x}} = \mathbf{x}_i - \mathbf{x}_j$ with $\mathbf{x}_i \neq \mathbf{x}_j$, then the Euclidean distance is [27]:

$$d = \|\mathbf{\Lambda}(\mathbf{x}_i - \mathbf{x}_j)\|_F^2,$$  (13)

where $\mathbf{\Lambda}$ is a diagonal matrix whose elements are singular, and $\| \cdot \|_F$ represents the norm.

Since the $\mathbf{u}_i, \mathbf{u}_j \in \mathbb{U}^{p \times 1}$ are the homologous vectors that have been modulated by the PAM according to Equation (12), we have:

$$d = \|\mathbf{\Lambda}\mathbf{S}(\mathbf{u}_i - \mathbf{u}_j)\|_F^2.$$  (14)

According to Equation (14), the minimum Euclidean distance is obtained as:

$$d_{\min}^i = \min_{j=1,\ldots\ldots,M^p, j\neq i} \|\mathbf{\Lambda}\mathbf{S}(\mathbf{u}_i - \mathbf{u}_j)\|_F^2,$$  (15)

where the $M$ means the size of the PAM constellation. Since the $\mathbb{U}$ have $p$ symbols, thus the total number of the transmitted vectors **u** is $M^p$.

To improve the reliability, the minimum distance should be maximized [26]. Then we propose to use the symmetric matrix **S** given by

$$\text{find}: \ \mathbf{S}$$
$$\text{maximize}: \min_{j=1,\ldots\ldots,M^p, j\neq i} \|\mathbf{\Lambda}\mathbf{S}(\mathbf{u}_i - \mathbf{u}_j)\|_F^2,$$  (16)

where in Equations (15) and (16), we could use the traversal method to select the optimal **S**, which in each transmission period is updated with $\mathbf{\Lambda}$.

### 3.3. Receiver Structure

The receiver structure and the decoding process of the received signal are shown in Figure 3. At the $j$th user's receiver, the received signals $\mathbf{y}_j$ [15] is

$$\mathbf{y}_j = \mathbf{H}_j \mathbf{s} + \mathbf{n}_j,$$  (17)

where $\mathbf{n}_j$ represents the additive white Gaussian noise (AWGN) vector [26].

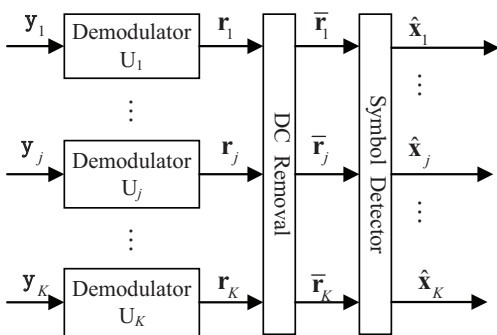

**Figure 3.** The receiving structure of the structure.

From Figure 3, we can see that the received signals can be attained from the equation $\mathbf{r}_j = \mathbf{U}_j^T \mathbf{y}_j$. Due to the orthogonal properties, $\mathbf{H}_j$ and $\widetilde{\mathbf{V}}_i^{(0)}$ satisfy the following condition $\mathbf{H}_j \cdot \widetilde{\mathbf{V}}_i^{(0)} = \mathbf{0}, i \neq j$. When the DC bias is removed [26], we can obtain

$$\bar{\mathbf{r}}_j = \mathbf{\Lambda}_j \mathbf{x}_j + \mathbf{U}_j^T \mathbf{n}_j = \mathbf{\Lambda}_j \mathbf{S}_j \mathbf{u}_j + \mathbf{U}_j^T \mathbf{n}_j. \tag{18}$$

Then we use the maximum likelihood (ML) detection method to recover the signal $\mathbf{x}_j$ of the user $j$ [26] as

$$\widehat{\mathbf{x}}_j = \arg\max_{\mathbf{x}_j} p(\bar{\mathbf{r}}_j | \mathbf{x}_j, \mathbf{\Lambda}_j) = \arg\min_{\mathbf{x}_j} \|\bar{\mathbf{r}}_j - \mathbf{\Lambda}_j \mathbf{x}_j\|_F^2, \tag{19}$$

where $\widehat{\mathbf{x}}_j$ that will be transmitted to the user $j$ is the estimates of $\mathbf{x}_j$.

*3.4. Theoretical SER Analysis*

In this part, we analyze the theoretical symbol error rate (SER) of the proposed scheme. First, from the ML detection in the receiver, we can define the conditional pairwise error probability (PEP) as the detector mistaking $\mathbf{x}_i$ into $\mathbf{x}_j$. Then we can receive the conditional PEP as [18]:

$$PEP(\mathbf{x}_i \longrightarrow \mathbf{x}_j | \mathbf{\Lambda}) = Q\left(\sqrt{\frac{\Gamma \|\mathbf{\Lambda}(\mathbf{x}_i - \mathbf{x}_j)\|_F^2}{4}}\right), \tag{20}$$

where the $\Gamma$ means the signal-to-noise ratio (SNR).

Then we apply the minimum Euclidean distance to Formula (20), and average over all the conditional PEP [25], we can have:

$$PEP(e|\mathbf{\Lambda}) = \frac{1}{M^p} \sum_{i=1}^{M^p} Q\left(\sqrt{\frac{\Gamma \cdot \min_{i \neq j} \|\mathbf{\Lambda}\mathbf{S}(\mathbf{u}_i - \mathbf{u}_j)\|_F^2}{4}}\right). \tag{21}$$

Finally, we numerically approximate the integral sum of Formula (21) to obtain a lower bound of the SER for the proposed scheme:

$$P_{e,l} \approx \frac{1}{W} \sum_{w=1}^{W} PEP(e|\mathbf{\Lambda}), \tag{22}$$

where $W$ is the number of the samples and $w$ is the index of the sample [25].

## 4. Numerical Results

In this part, we first present the simulation parameter settings. Then we provide simulation results to verify the effectiveness of the proposed scheme.

In the simulation, we assume that there are $K = 2$ users, and each user is equipped with $N_t = 2$ transmit apertures and $N_r = 2$ receiving apertures. We assume that a laser with a beam width of 3 mm and a beam wavelength of 550 nm is used as the light source. We set the photoelectric conversion efficiency $\eta$ to be 1 and the field of view (FOV) semi-angle of the receiver to be $\Psi_{1/2} = 45°$. The receiver aperture diameter is set as $D = 0.05$ m and the divergence semi-angle of transmitters is set as $\Phi_{1/2} = 0.5°$. The distance between receivers is set to $s_r = 0.25$ m, while the distance between transmitters is set to $s_t = 0.25$ m.

First of all, we set $C_n^2 = 1 \times 10^{-14}$, which means that the turbulence is weak. Additionally, we set the link distance in the atmosphere at 500 m and the distance underwater at 200 m, which can be obtained in the air channel $\sigma_I^2 = 0.18$ and in the underwater channel $\sigma_I^2 = 0.03$. From Figures 4 and 5, we can see that the theoretical results match the simulation ones. Moreover, we can see that the proposed scheme outperforms the other three benchmark schemes. As we can see in Figure 4, the proposed symmetric scheme outperforms the SMP scheme [14,15] by about 4 dB at the SER = $1 \times 10^{-4}$ level in the atmospheric channel, while in the underwater channel, the proposed symmetric scheme obtains a performance gain of about 3 dB at SER = $1 \times 10^{-4}$ comparing to the SMP scheme [14,15] in Figure 5. This also means the proposed symmetric scheme has a good advantage of improving reliability.

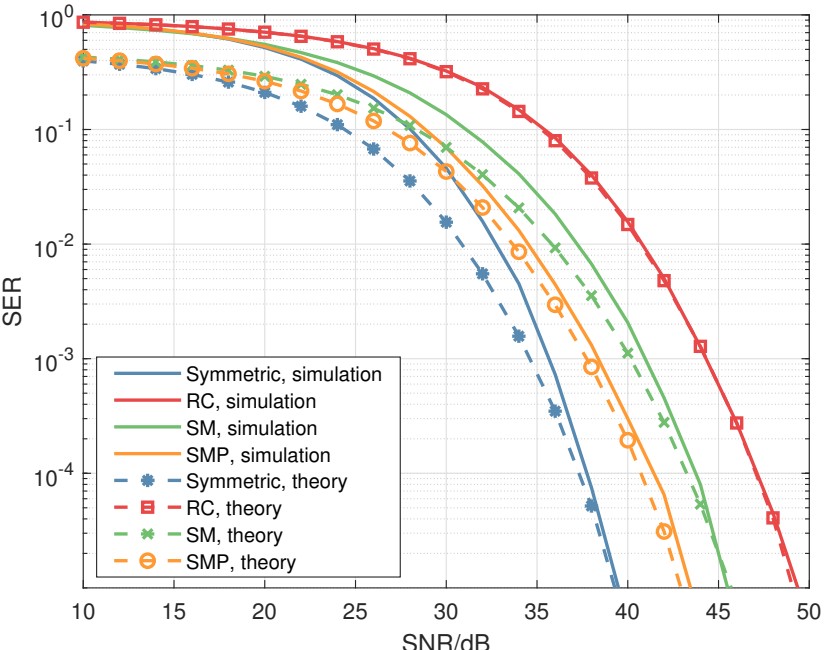

**Figure 4.** Theoretical and simulation results in the atmospheric channel, $\sigma_I^2 = 0.18$.

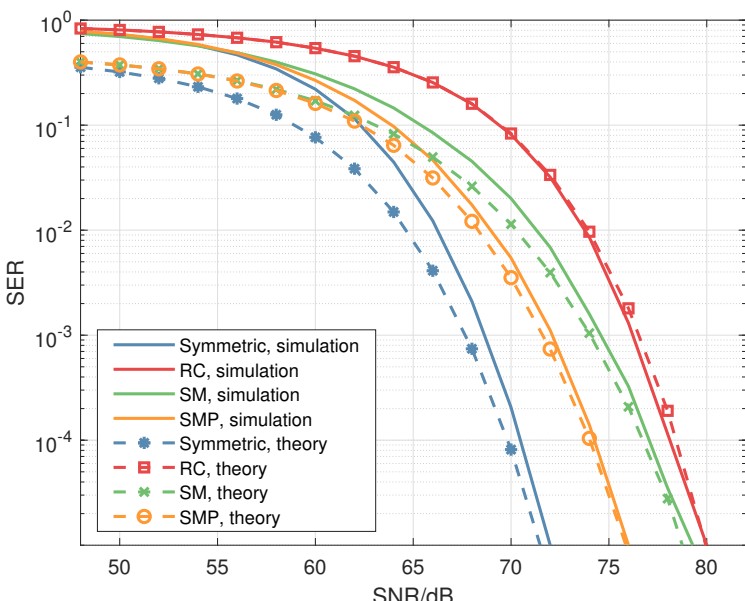

**Figure 5.** Theoretical and simulation results in the underwater channel, $\sigma_I^2 = 0.03$.

Next, we investigate the reliability performance over channels with different turbulence intensities. In the simulations, we set the distance in the atmosphere and underwater as 500 m and 200 m, respectively. For the atmospheric channel, we set $\sigma_I^2 = 0.18$ and $\sigma_I^2 = 1.68$, respectively, to consider the weak turbulence and strong turbulence effects, while for the underwater channel, we set $\sigma_I^2 = 0.03$ and $\sigma_I^2 = 1.39$ to, respectively, describe the weak and strong turbulence. It can be seen from Figures 6 and 7 that the reliability performance of our design is better than the other three benchmark schemes. In Figure 6, the channel is in the atmosphere, when the SER is $1 \times 10^{-5}$ level, the symmetric scheme is about 4 dB better than the SMP scheme [14,15] for $\sigma_I^2 = 0.18$ and about 3 dB better than the RC scheme [14,15] for $\sigma_I^2 = 1.68$. Additionally, it can be seen in Figure 7 that when the channel is the underwater, at the $1 \times 10^{-3}$ level, the proposed scheme outperforms the SMP scheme [14,15] by about 5 dB for $\sigma_I^2 = 0.03$ and has a performance gain of about 7 dB over the RC scheme [14,15] for $\sigma_I^2 = 1.39$. Furthermore, we can see that the proposed scheme is better than the other three schemes in improving the reliability of communication in both weak and moderate to strong turbulence situations.

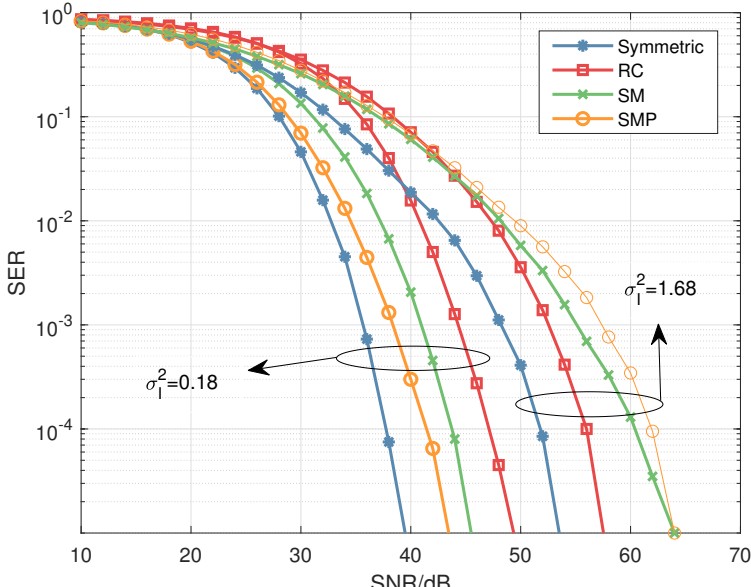

**Figure 6.** Reliability performance comparisons for the atmospheric channel.

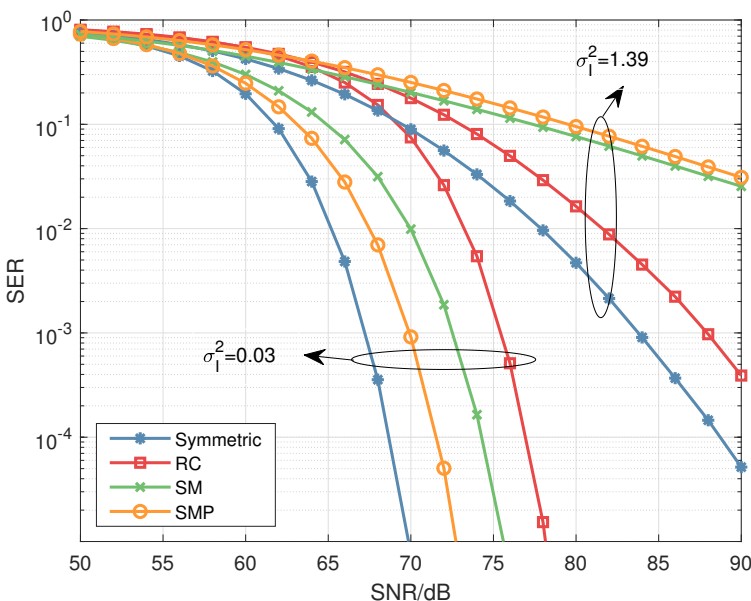

**Figure 7.** Reliability performance comparisons for the underwater channel.

Here we study the reliability performance of the four schemes for the air–water interface channel. In the simulation, we set the channel parameters as follows: wind speed is 1 m/s, atmospheric propagation distance is 100 m, and underwater propagation distance is 20 m. As can be seen from Figure 8, the proposed symmetric scheme is still superior to the other benchmark schemes in improving the reliability performance. It has a good performance gain of about 7 dB comparing to the RC scheme [14,15] when the SER is $1 \times 10^{-3}$.

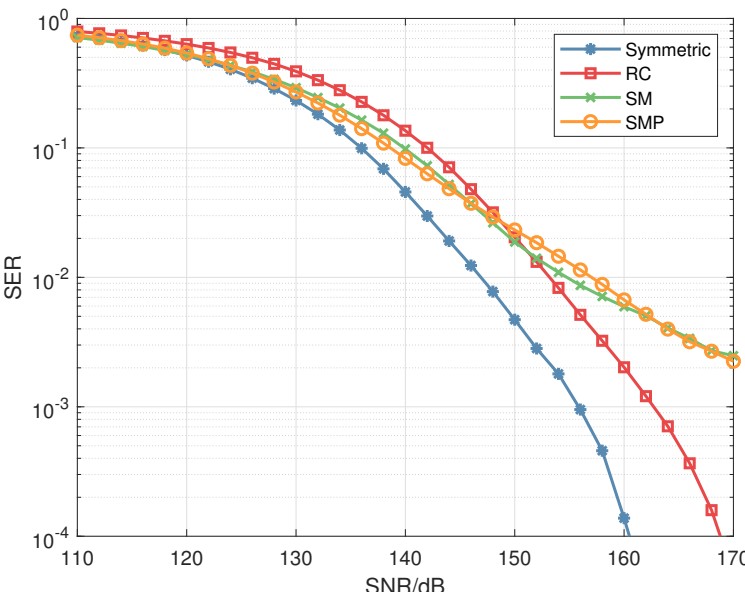

**Figure 8.** Reliability performance comparisons for the air–water interface channel.

Subsequently, we investigate the reliability performance with different transmission distances of 500 m, 800 m, 1100 m, and 1400 m, respectively, in the atmosphere in Figure 9. We can notice that the reliability performance will become better when the transmission distance is shorter. We can see that at the $1 \times 10^{-5}$ level, when the distance is 500 m, it outperforms by 6 dB comapred to when the distance is 800 m, it has a performance gain of about 10 dB compared to when the distance is 1100 m, and is about 20 dB better than when the distance is 1400 m.

Then, we studied the reliability of our proposed scheme underwater with different water types. Clear water, coastal water, and harbor water are considered. From Figure 10, we can see that in clear water it has a good performance gain of about 30 dB compared to when the water type is coastal water and about 98 dB when the water type is harbor. The transmission in the clear water is very good because the impurities and other small particles in the clear water are the least, and in the harbor water they are the greatest. Without these adverse factors that affect the communication propagation process, light propagation underwater is more convenient, so the reliability of the transmission will be better, while with these affecting the propagation, the propagation underwater is harder, so the reliability is worse.

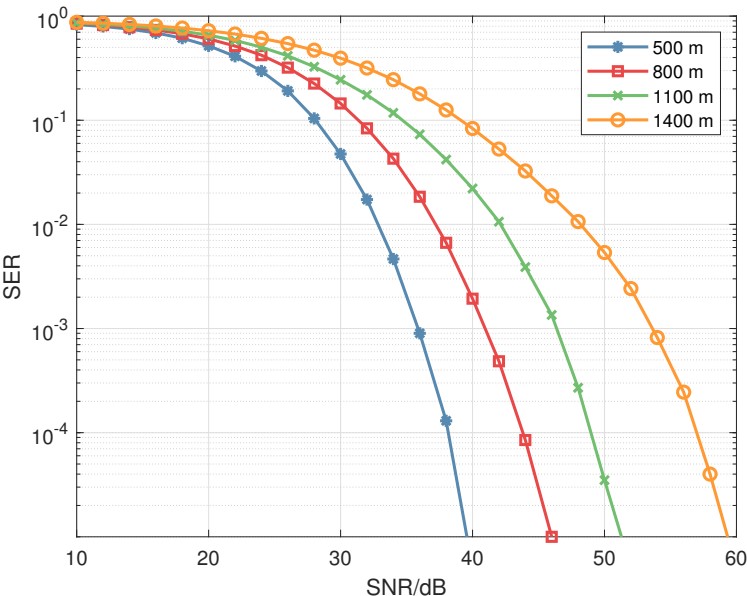

**Figure 9.** Comparisons with different link distances for the atmospheric channel.

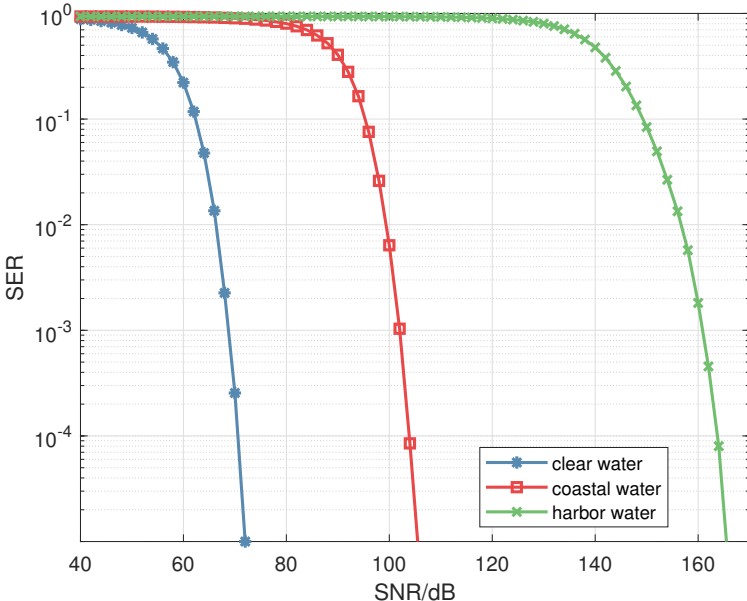

**Figure 10.** Comparisons for the underwater channel when the water type is different.

Finally, for the air–water interface channel, we study the reliability performance of our proposed scheme with different wind speeds. In the simulation, we set the link distance as 100 m in the atmosphere and 20 m in the water. Moreover, we set the wind speed as 0 m/s,

1 m/s, and 4 m/s, respectively. Then we can attain the simulation in Figure 11, we can see that the higher the wind speed, the worse the reliability performance. When the SER is at the $1 \times 10^{-5}$ level, when the wind speed is 0 m/s, it has a performance gain of about 6 dB compared to when the wind speed is 1 m/s, and about 9 dB compared to when the wind speed is 4 m/s. This is because the higher the wind speed, the more horizontal the plane bends, and the more distorted the light propagation on the horizontal plane, resulting in fewer photons being able to travel to the receiver. Therefore, the reliability of transmission is degraded.

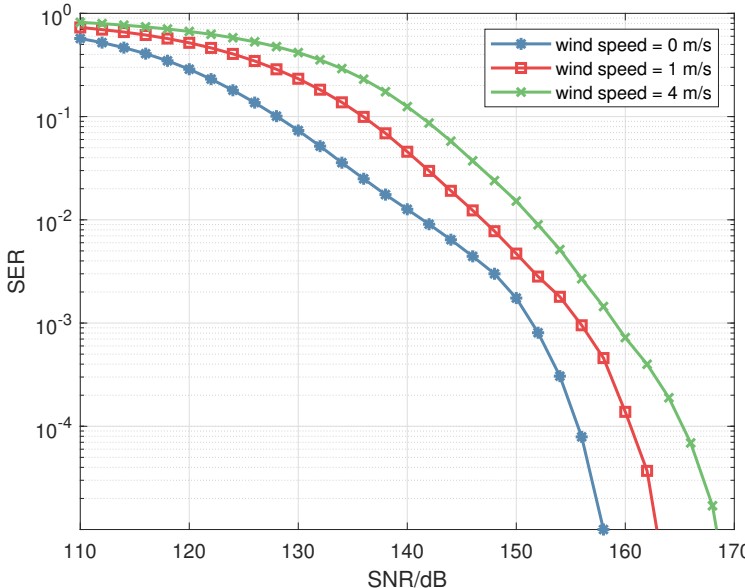

**Figure 11.** Comparisons with different wind speeds for the air–water interface channel.

## 5. Conclusions

In this paper, we propose a symmetric matrix-assisted MIMO-VLC system for maritime communication to improve reliability performance. By exploiting the repeatability of the elements of the symmetric matrix, the Euclidean distance can be enlarged to combat the adverse conditions of channels and thus the reliability performances can be improved. Then we derive the theoretical SER expressions of the proposed scheme. Moreover, simulations are conducted to verify the effectiveness of the proposed scheme and to validate that the proposed scheme is superior to the benchmark schemes over atmospheric channels, underwater channels, and air–water interface channels. Then we simulate the proposed scheme over atmospheric channels, underwater channels, and air–water interface channels with different channel parameters. These simulation results show that the proposed symmetric matrix scheme has advantages in improving communication reliability under the three channels. Moreover, when changing the environmental parameters, we can observe that the shorter the propagation distance, the cleaner the water and the lower the wind speed, the transmission reliability can be improved. For atmosphere channel, when the SER is $1 \times 10^{-5}$, the symmetric scheme is about 4 dB better than the SMP scheme when $\sigma_I^2 = 0.18$ and about 3 dB better than the RC scheme when $\sigma_I^2 = 1.68$. Therefore, the symmetric matrix-aided MIMO transmission scheme provides a promising solution for reliable transmission. Further research directions on improving the reliability performance include applying the geometric mean decomposition (GMD) or combining the SVD with the GMD for enhancing the reliability performance.

**Author Contributions:** Software, Y.Z.; investigation, Y.F.; writing—review and editing, L.Z.; project administration, Z.W. All authors have read and agreed to the published version of the manuscript.

**Funding:** This research received no external funding.

**Institutional Review Board Statement:** Not applicable.

**Informed Consent Statement:** Not applicable.

**Data Availability Statement:** All data is available in the manuscript.

**Acknowledgments:** This work was supported by the Project of National Natural Science Foundation of China under Grant U20A20162.

**Conflicts of Interest:** The authors declare no conflict of interest.

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
