# Peer review of "A Symmetric Matrix-Aided MIMO to Improve Reliability for Maritime Visible Light Communications"

_electronics, doi:10.3390/electronics11193019_

Round 1

Reviewer 1 Report

The paper is well written and understandable for expert and non-expert audiences. The contribution of presenting and evaluating a MIMO-VLC system for maritime environments is understood. However, I have certain doubts and suggestions that, if they are not well resolved, could be critical at work.

1) Mathematically speaking, how does the dual SVD method eliminate or mitigate multi-user interference? Please explain it.

2) Strictly speaking, equations 4 and 5 represent the lognormal probability density functions for the channel coefficient corresponding to turbulence. In the article, in my opinion, these are not well stated or derived statistically. Please present the derivation of these expressions, since if they are not very explicit, the article would lose solidity and validity.

3) The conclusions are very basic and do not refer to hard data obtained on the results. Please, I suggest rewriting them and presenting them considering details of some relevant data obtained.

Reviewer 2 Report

1. Authors should few lines on illustration and comparison the results in context to SNR etc and outline the importance of research work carried out.

2. Presently, it is hard to read data from figures hence, Authors should redraw all the figures and ensure that the text and numerical data in the figures are readable and figures are of high quality. 

3. In the conclusion section, the authors should add a few sentences on the significance and future directions of the results obtained and compare them to the existing literature using SNR/dB.

Round 2

Reviewer 1 Report

Thanks to the authors for the clarifications and corrections for the improvement of the article. The paper can be published in its current state.